# An Improved Speciation Method Combining IC with ICPOES and Its Application to Iodide and Iodate Diffusion Behavior in Compacted Bentonite Clay

**DOI:** 10.3390/ma14227056

**Published:** 2021-11-20

**Authors:** Chuan-Pin Lee, Yanqin Hu, Dongyang Chen, Enhui Wu, Ziteng Wang, Zijin Wen, Neng-Chuan Tien, Fan Yang, Shih-Chin Tsai, Yunfeng Shi, Yi-Ling Liu

**Affiliations:** 1School of Nuclear Sciences and Engineering, East China University of Technology, Nanchang 330013, China; bennis6723@139.com (C.-P.L.); hyqdrx@163.com (Y.H.); chen_dongyang2021@163.com (D.C.); 18679656079@139.com (E.W.); wzt2844@163.com (Z.W.); 201960370@ecut.edu.cn (Z.W.); syf541006935@126.com (Y.S.); 2Nuclear Science and Technology Development Center, National Tsing Hua University, Hsinchu 300044, Taiwan; sctsai@mx.nthu.edu.tw; 3Xiamen Institute of Rare Earth Materials, Haixi Research Institute, Chinese Academy of Sciences, Xiamen 361021, China; 4Department of Nuclear Environmental Science, China Institute for Radiation Protection (CIRP), Taiyuan 030006, China; 5School of Political Science and Public Administration, Huaqiao University, Fujian 362021, China; helenyilingliu@hotmail.com

**Keywords:** iodide (I^−^), iodate (IO_3_^−^), IC-ICPOES, bentonite, effective diffusion coefficient, accessible porosity, anion exclusion effect

## Abstract

An accurate and effective method combining ion chromatography (IC) and inductively coupled plasma optical emission spectrometry (ICP-OES) was applied in this work to qualitatively and quantitatively analyze individual and co-existing iodide (I^−^) and iodate (IO_3_^−^) at various concentrations. More specifically, a very strong linear relationship for the peak area for the co-existing I^−^ and IO_3_^−^ ions was reached, and a high resolution value between two peaks was observed, which proves the effectiveness of our combined IC-ICP-OES method at analyzing iodine species. We observed lower accessible porosity for the diffusion of both I^−^ and IO_3_^−^ in samples of bentonite clay using IC-ICP-OES detection methods, where the effective diffusion coefficient varied based on the anion exclusion effect and the size of the diffusing molecules. In fact, the distribution coefficients (*K_d_*) of both I^−^ and IO_3_^−^ were close to 0, which indicates that there was no adsorption on bentonite clay. This finding can be explained by the fact that no change in speciation took place during the diffusion of I^−^ and IO_3_^−^ ions in bentonite clay. Our IC-ICP-OES method can be used to estimate the diffusion coefficients of various iodine species in natural environments.

## 1. Introduction

The design of a deep geological repository for spent fuel or high-level radioactive waste (HLW) generally adopts a “multi-barrier system”. Generally, the underground facilities, waste containers, and buffer/backfill materials (e.g., bentonite clay) are known as engineering barriers, and the surrounding geological formations are known as natural barriers. In such a system, the engineering barriers and the geological medium serve a dual purpose: they ensure both the containment of the source term and the protection of the biosphere. In order to retard radionuclide release, the materials used in the construction of an HLW repository must have specific properties that help to prevent the dispersion of radioactivity. Such design parameters make it possible both to delay the time required for the radionuclides to reach the biosphere and to reduce their concentration by means of retarding mechanisms, including sorption, diffusion, and dispersion, so as to ensure the safety of the final disposal of radioactive waste [1,2].

Bentonite clay is just such an engineering barrier that has often been used as a buffer/backfill material because of its retention effect on the migration of radionuclides to the biosphere [2,3]. Conducting performance assessments (PA) of Wyoming bentonite clay and other such materials in order to determine their physical and chemical properties has been a key element in the design of HLW repositories in the past 20 years. Studies have used experimental methods to identify the properties of bentonite, such as low hydraulic conductivity, good swelling and plasticity, low solute diffusivity, strong retardation of radionuclide migration, and high cation exchange capacity [4,5,6,7,8].

In addition to transuranic elements (TRU), the possible environmental impact of long-lived radionuclides that exhibit a relatively high mobility under certain geochemical conditions must be taken into consideration when designing an HLW repository. In particular, weakly adsorbed radionuclide anions are the most important, for example ^129^I, ^99^Tc, and ^79^Se [9,10,11,12]. As one of the primary fission products, ^129^I is characterized by a long half-life (t_1/2_ = 1.57 × 10^7^ years), a high fission yield, easy volatilization, easy migration, a high radioactive toxicity, and a high bioavailability. Furthermore, the pH of and the redox potential in aqueous solutions determines the form that iodine takes in the environment, including iodide (I^−^), iodate (IO_3_^−^), I_2_, and organic iodine. Thus, ^129^I is the main source of potential risk posed, not only by HLW repositories, but also by nuclear accidents such as the 2011 Fukushima Daiichi nuclear disaster. This threat has recently been made obvious when ^129^I was observed in the Pacific Ocean around Japan, where fast dispersion by the movement of the seawater and deposition caused it to migrate into the marine sediments [13].

While carrying out a safety assessment (SA) for the disposal of HLW, numerical models should be used in order to consider and evaluate a variety of scenarios, including the possibility of radionuclides being released into the environment and thus exposing people to radiation, as in the scenario in which radionuclides migrate into groundwater following the failure of the multi-barrier system after 10,000 years [14]. In fact, groundwater is the main carrier of migrating ^129^I, which can exist in various forms, of which I^−^ and IO_3_^−^are the main anions found in groundwater [15,16]. Therefore, the diffusion behavior of I^−^ and IO_3_^−^ in the bentonite clay buffer is considered a major issue for the assessment of the possibility of the release of radionuclides into the environment from an HLW repository.

Several previous studies on fission products have been performed to calculate sorption and diffusion parameters, *K_d_* values, and diffusion coefficients under various conditions. For example, some research has recorded very low *K_d_* values (~0) [17,18]. Another study in which through-diffusion experiments were performed with sand with bentonite clay mixtures obtained diffusion coefficients of ^125^I^−^: 5.1 ± 0.8 × 10^−11^ m^2^ s^−1^ [19]. Moreover, it has been suggested that the migration of iodine may be affected by its own physical and chemical properties, as in the case of the anion exclusion effect [19,20]. Finally, it is also possible that oxidation-reduction reactions occurring in the environment cause changes in the chemical form of iodine, thus also impacting its migration. However, few studies have focused on iodine speciation (e.g., I^−^ and IO_3_^−^), on the one hand, because of the difficulty of separating the iodine from the environmental matrix and, on the other, because of the high cost of achieving accurate detection with a combination of advanced instrumental methods such as accelerator mass spectrometry (AMS), nanoscale secondary ion mass spectrometry (NanoSIMS), and inductively coupled plasma mass spectrometry (ICP-MS) [13,16,21,22].

In recent years, ion chromatography (IC) has become a popular and effective method applied to the analysis of anion speciation in environmental samples [16]. Another method used for quantifying anions based on deriving analytical solutions is inductively coupled plasma optical emission spectrometry (ICP-OES), a convenient and rapid tool with which the limit of detection can be determined. Therefore, in this paper, we propose a relatively simple, effective, and reliable method to detect I^−^ and IO_3_^−^ by combining IC and ICP-OES to create IC-ICP-OES. More specifically, we investigated the variability of the concentrations of I^−^ and IO_3_^−^, both as individual ions and as co-existing ions, separating and analyzing them by calibrating the IC-ICP-OES to be able to carry out both qualitative and quantitative analyses. We also performed through-diffusion experiments to examine the diffusion behavior of I^−^ and IO_3_^−^ in compacted bentonite clay based on their diffusion coefficients. Our experiments with the IC-ICP-OES can serve as an important reference case while carrying out future SAs of engineering barriers used in HLW repositories in China.

## 2. Materials and Methods

### 2.1. Characterization of Bentonite Clay

The material tested in this work was a sample of Wyoming Na-bentonite clay purchased from American Colloid Company. It had a montmorillonite content of approximately 85–90 wt.%. Previous studies have investigated its mineralogical and chemical properties using X-ray diffraction analysis techniques to identify several minor mineral components such as quartz and feldspar [23,24]. In this study, a wavelength dispersive X-ray fluorescence (WDXRF) spectrometer (Axios, PANalytical Inc., Amsterdam, the Netherlands) was employed to analyze samples of commercial bentonite clay from Wyoming (B) and Taiwan (Z), as well as a sample of standard montmorillonite from Inner Mongolia in China (M). The chemical compositions of these clays are given in Table 1. These results were compared for reference.

### 2.2. Preparation of Iodide (I^−^) and Iodate (IO_3_^−^) Standard Solution

All chemical reagents used in our experiments were of analytical grade of purity, and the Milli-Q^®^ ultrapure water has a resistivity of 18.2 MΩ.cm (Merck, Darmstadt, Germany). A standardized potassium iodide (KI) solution containing 1.20 ± 0.05 g of KI and a potassium iodate (KIO_3_) solution containing 1.30 ± 0.05 g of KIO_3_ were prepared in deionized water and 3–5% of nitric acid (HNO_3_). Finally, these solutions were diluted 100 times with deionized water to 1 L in a volumetric flask. The concentrations of I^−^ and IO_3_^−^ in the diluted solutions were measured and analyzed by means of IC and ICP-OES (iCAP 7000, Thermo Fisher Scientific Inc., Waltham, MA, USA) to the point at which the variation of element concentrations was within 5% of the corresponding liquid phase concentrations.

### 2.3. Qualitative and Quantitative Analyses with IC-ICP-OES to Detect I^−^ and IO_3_^−^

We used an IC instrument equipped with a conductivity detector to analyze the different iodine species, and then we quantitatively measured them with ICP-OES. By deriving an analytical solution, the I^−^ and IO_3_^−^ anions could be effectively detected and separated through the combination of IC and ICP-OES techniques. In addition, an ICS600 system (Thermo Fisher Scientific Inc., Waltham, MA, USA) was equipped with a gradient pump, an injection valve with a 10 μL sample loop, and an anion-exchange column (250 mm × 4.6 mm inner diameter). Moreover, the eluent solution used in the IC procedure consisted of 1.0 mM NaHCO_3_ and 3.2 mM Na_2_CO_3_, with degassing performed with a DG-2410 degasser (Uniflows, Tokyo, Japan) before the injection of individual and co-existing I^−^ and IO_3_^−^ ions. Finally, the resolution (R) was calculated based on the peak areas and retention times:R = 0.5 × (M_1_ + M_2_)/(M_1_ − M_2_) × (s/a)(1)
where M_1_ and M_2_ represent the peak areas of I^−^ and IO_3_^−^, respectively; a is the peak width at 5% of the peak height (for I^−^ and IO_3_^−^); and s is the time difference between the peak heights of I^−^ and IO_3_^−^. The results showed that two peaks could be effectively separated and identified at R > 0.5.

### 2.4. Through-Diffusion Experiments (TD): Column Tests

We developed and verified a precise and reliable apparatus with sandwich-like columns to conduct through-diffusion (TD) experiments, as shown in Figure 1. A similar experimental apparatus has been employed in several past studies [12,20,25]. The MX-80 bentonite sample dimension in thickness and cross-sectional area was 3 mm and 19.6 cm^2^, respectively. The sample slab was inserted and fixed in a holder and was sandwiched between two glass microfiber filters (0.4 mm thickness) with pore size of 0.7 um (GF/F, Associated Design & Manufacturing Co., Alexandria, VA, USA), a Teflon O-ring, and constrainers. Figure 1 shows a schematic description of the diffusion cell for compacted bentonite in the through-diffusion experiments. The Teflon O-ring and constrainers were used to reduce cell deformation, minimize diffusion resistance, and resist swelling pressure induced by the bentonite clay following saturation with water. The previous literature reported that concentration gradient between inlet and outlet diffusion cells with filters has few differences in estimating effective diffusion coefficients of HTO, ^125^I^−^ and ^134^Cs^+^ [19].

The setup consisted mainly of a highly precise multichannel isocratic peristaltic pump (Masterflex L/S, Cole-Parmer Instrument Co., Barrington, IL, USA) for use with two polypropylene (PP) columns (No. 1–2, modified type, Hsinchu ZeGi Industrial Co., Ltd., Hsinchu, Taiwan), an initial tracer reservoir containing individual I^−^ (C_0_ = 1000 ± 50 ppm) and IO_3_^−^ (C_0_ = 1000 ± 50 ppm) ions, and various Teflon^®^ (PTFE) units and connectors. The PP columns were made of pressure-resistant polypropylene (<10 MPa) and had a length of 13.6 cm and an inner diameter of 5 cm. They were filled with bentonite clay having a total porosity of 0.26 and a bulk density of 2.0 g/cm^3^.

### 2.5. Estimation of Diffusion Coefficient

In this work, Fick’s second law was applied to estimate the diffusion coefficients (*D*) for I^−^ and IO_3_^−^ in compacted bentonite clay. The one-dimensional diffusion equation for the radionuclides can be written as follows:(2)$$\frac{\partial C}{\partial t}=\frac{D_{e}}{\alpha }\frac{\partial^{2}C}{\partial x^{2}},\;\left(D_{a}=\frac{D_{e}}{\alpha },\alpha =\theta +\rho_{b}K_{d}\right)$$
where *D_e_* and *D_a_* are the effective and apparent diffusion coefficients, respectively; α (capacity factor) is a function of the bulk density of the dry material (*ρ_b_*), the total porosity of compacted samples (*θ*), and the distribution coefficient *K_d_*; and C is the concentration of the solute in the liquid phase. The rate at which molecules spread determines the diffusion coefficient (*D*), which is the proportionality constant between the mass flux and the concentration gradient of the solute. Both boundary and initial conditions limit the through-diffusion experimental method and can be expressed as follows:C(x,0)=0, 0<x<L
$$C\left(0,t\right)=C_{0}$$
C(L,t)~0
where the tracer concentration in the reservoir containing the tracer is constant (*C*_0_); the ion concentration in the opposing reservoir is kept close to zero; and *L* is the overall duration of the compaction of the clay samples.

As defined by Crank [26], the cumulative mass, M, and the concentration ratio, CR(t), of I^−^ and IO_3_^−^ in the measurement reservoir were calculated according to the following formula:(3)$$\mathrm{CR}(t)=\frac{\sum C\left(t\right)}{C_{0}}=\frac{LS}{V}\left(\frac{D_{e}t}{L^{2}}-\frac{\alpha }{6}-\frac{2}{\pi^{2}}\sum_{n=1}^{\infty }\frac{{\left(-1\right)}^{n}}{n^{2}}\exp\left[\frac{-n^{2}\pi^{2}D_{e}t}{L^{2}}\right]\right)$$
where *D_e_* is the effective diffusion coefficient that takes the capacity factor (α) into consideration and V and S are the diffusion volume and the cross-section area of the compacted samples, respectively. When sufficient time is allowed to carry out the TD test, the diffusion process reaches a steady state. Consequently, the exponential term in Equation (3) tends to zero, and the profile curve illustrating the distribution of the CR(t) shows a linear relationship with time (*t*). Moreover, using a method for statistical data analysis that had previously been developed [25], we compared our measurements for I^−^ and IO_3_^−^ and evaluated them in terms of the steady-state diffusion condition and its relation to time. The combination of the IC and ICP-OES techniques makes it possible to determine whether or not the speciation of I^−^ and IO_3_^−^ changes during the diffusion process. Therefore, using both techniques while concurrently performing experimental and numerical analyses of I^−^ and IO_3_^−^ provides an effective and important tool for carrying out future safety assessments of HLW repositories.

### 2.6. Estimation of Distribution Coefficients (K_d_)

We performed a simulation of batch sorption experiments with I^−^ and IO_3_^−^ in bentonite clay. More specifically, the standard ASTM batch method [27] was applied to three solutions of 50 mL, each containing samples of bentonite prepared in centrifuge tubes. Thus, the adsorption of I^−^ and IO_3_^−^ in the bentonite was observed by measuring the pH, Eh, and the final ion concentrations. In order to introduce stable isotope tracers, an individual I^−^ and IO_3_^−^ stock solution was added to the initial bentonite solution prior to the batch testing. The I^−^ and IO_3_^−^ stock solution was prepared at initial concentrations (*C*_0_) of 5, 10, and 20 ppm. All batch tests were conducted on solutions with a solid–liquid ratio of 1 g per 30 mL. The centrifuge tubes were placed in an oscillating thermostatic shaker, and the solutions were uniformly mixed at 200 rpm for 7 days. The solid and liquid phases were separated by being put in a high-speed centrifuge (RCF = 10,000 g) for 30 min. Approximately 10 mL of the supernatant was reserved to analyze the final concentrations of the I^−^ and IO_3_^−^ by means of a combination of the IC and off-line ICP-OES techniques, at which time changes in pH and Eh were also recorded. As a result, the distribution coefficient (*K_d_*) for I^−^ and IO_3_^−^ can be written as follows:(4)$$K_{d}=\frac{C_{0}-C}{C}\frac{V}{m}$$
where *C*_0_ is the initial ion concentration of I^−^ (or IO_3_^−^); C is the final ion concentration; and V and m are the volume (mL) and mass (g) of the samples, respectively.

## 3. Results

### 3.1. Identification for Individual I^−^ and IO_3_^−^

Solutions (10 μL) containing various concentrations of I^−^ and IO_3_^−^ ions were injected into an IC system, with measurements simultaneously taken by a conductivity detector and ICP-OES equipment and elution performed with a solution of 1.0 mM NaHCO_3_ and 3.2 mM Na_2_CO_3_ (See results in Table 2). After diluting the solutions 100–500 times, distinct peaks at 20 min and 4 min on the chromatograms for the individual I^−^ and IO_3_^−^, respectively, with a very clear peak for IO_3_^−^, show that these ions were effectively separated and detected by IC in the range 100–500 ppm (See Figure 2). This finding also demonstrates that ICP-OES was able to identify the I^−^ and IO_3_^−^ ions as forms of iodine after linear calibration had been performed. However, conductivity peaks were much less prominent at the concentration of 100 ppm, which made it very difficult to distinguish I^−^ and IO_3_^−^.

### 3.2. Performance for Co-Existing I^−^ and IO_3_^−^

IC was also used to analyze solutions with different concentrations of co-existing I^−^ and IO_3_^−^ ions (Table 3 and Figure 3). The resulting chromatograms are in agreement with those shown in Figure 2 in terms of individual peak height, peak area, and retention time. Moreover, a very strong linear relationship for the peak area for the co-existing I^−^ and IO_3_^−^ ions was reached (0.99), and a high resolution value (R: 52.30~79.36) between two peaks was observed, which proves that IC provided a very effective qualitative and quantitative analysis of the two forms of iodine investigated in this study.

### 3.3. Diffusion of I^−^ and IO_3_^−^

The results of the TD experiments with our IC-ICP-OES method indicate that the time required for the I^−^ and IO_3_^−^ ions to diffuse out of the bentonite clay was approximately 24 h, with the steady-state condition reached in 7 to 10 days as a result of a constant diffusion flux. The effective diffusion coefficient (*D_e_*) and the capacity factor (α) for the I^−^ and IO_3_^−^ ions in compacted bentonite were estimated on the basis of numerical analyses, including calculating graphical asymptotes, finding analytical solutions with Lsqcurvefit, and modeling with dual period diffusion model (DPDM) [25,28] (See Table 4 and Figure 4). Our findings indicate that the diffusion of the I^−^ and IO_3_^−^ ions in bentonite clay could be suitable fitting, which can be explained by the fact that no change in speciation was detected by means of the IC-ICP-OES technique.

### 3.4. Batch Tests of I^−^ and IO_3_^−^

The batch experiments conducted with the IC-ICP-OES method indicated that the adsorption of I^−^ and IO_3_^−^ in three samples of bentonite clay reached the steady-state condition in approximately 7 days. Table 5 lists the individual values of the pH, Eh, and distribution coefficients (*K_d_*) for the I^−^ and IO_3_^−^ ions in compacted bentonite. Our findings indicate that no sorption of I^−^ and IO_3_^−^ has taken place in the bentonite clay samples, which can be explained by the fact that no change in speciation was detected by means of the IC-ICP-OES technique.

## 4. Discussion

In this work, individual IO_3_^−^ ions were consistently detected first, after which individual I^−^ was extracted by elution from the solution containing co-existing I^−^ and IO_3_^−^ ions. Generally, the affinity of the anion exchange resin is quite different for I^−^ and IO_3_^−^, with I^−^ being subject to a higher adsorption affinity and anion exchange and showing a slower retention time (~20 min) than IO_3_^−^ (~4 min) during the elution process. In fact, it has been shown that some ions interfere with the IC-based detection of I^−^ and IO_3_^−^ ions in complex and natural environments [16,29], a problem that could be solved by combining IC with such instruments as ICP-MS, AMS, or other instruments for measuring radioactivity. To this end, we have developed a more effective and accurate technique based on the combination of the IC and the ICP-OES methods to analyze qualitatively and quantitatively both individual and co-existing I^−^ and IO_3_^−^ ions in solutions at various concentrations.

In this study, tritiated water (^3^H_2_O), a non-reactive radiotracer, was used to estimate the total porosity (θ = the ratio of pore volume to total volume) of the bentonite clay. Because the total porosity is equivalent to the water content, ^3^H_2_O is commonly applied to the identification of all chemical elements that are not retained in the solid phase. We showed that the porosity accessible (α_acc_) to both I^−^ and IO_3_^−^ ions was lower than that accessible to HTO, which was determined on the basis of the anion exclusion effect (*K_d_*~0) [20,25] and the decreasing number of interlayer pores [10,30,31]. This result indicates that I^−^ and IO_3_^−^ ions have a similar mobility as does ^3^H_2_O when the effective diffusion coefficients (*De*) are compared. Moreover, there were significant differences between the anions present in bentonite clay with regards to the exclusion effect, which has been associated with the size or hydrated radius of I^−^ (~2.16 Å), IO_3_^−^ (~3.30 Å), and ^99^TcO_4_^−^ (~3.50 Å) [32,33]. On the other hand, the porosity accessible (α_acc_) to SeO_3_^2−^ (IV) might be higher than that accessible to ^3^H_2_O as a result of the anion exclusion effect and the weak adsorption effect (*K_d_* > 0) [20]. The parameters of these different anions are listed in Table 4; the I^−^ and IO_3_^−^ anions were investigated in this paper and the ^99^TcO_4_^−^ and SeO_3_^2−^ (IV) anions were examined in previous research [20,25].

## 5. Conclusions

In this paper, we developed a relatively simple, effective, and reliable method to detect individual and co-existing I^−^ and IO_3_^−^ ions through a combination of ion chromatography (IC) and inductively coupled plasma optical emission spectrometry (ICP-OES) techniques, which we call IC-ICP-OES. The anion exchange resin exhibited a different affinity for I^−^ and IO_3_^−^, with I^−^ being subject to a higher adsorption affinity and anion exchange. This finding demonstrates that ICP-OES was able to identify the I^−^ and IO_3_^−^ ions as forms of iodine after linear calibration had been performed. In fact, there was obvious interference with the performance of the conductivity detector during the IC-based I^−^ and IO_3_^−^ analysis procedure, which suggested the possibility of combining IC with an offline ICP-OES detection system to obtain accurate measurements of I^−^ and IO_3_^−^ chromatographic peaks at different retention times. Therefore, an analysis of the diffusion of I^−^ and IO_3_^−^ ions in bentonite clay was performed, with the results showing that lower accessible porosity was related to the anion exclusion effect and the size or hydrated radius of the ions. The effective diffusion coefficient (*D_e_*) and the capacity factor (α) obtained in this study could be used as input data for future safety assessments of radioactive waste disposal facilities. In conclusion, ^129^I anions, in the form of I^−^ and IO_3_^−^, are radionuclides that are critical factors to take into account when carrying out safety assessments of deep geological repositories for HLW, which should be a focus of future R&D studies. In addition, our results confirm that Wyoming bentonite clay shows potential for use as a buffer/backfill material to counter the diffusion of radionuclides from containers used for storing HLW.

We have been relying on nuclear power for decades to meet our electricity needs. Today, governments around the word continue to invest in nuclear technology as part of their strategies to combat climate change and transition to low-carbon economies. Our role is to ensure there is a safe, long-term solution to the obstacles facing the containment and isolation of any spent nuclear fuel created by nuclear power generation, including what already exists today and any produced in the future.

## Figures and Tables

**Figure 1 materials-14-07056-f001:**
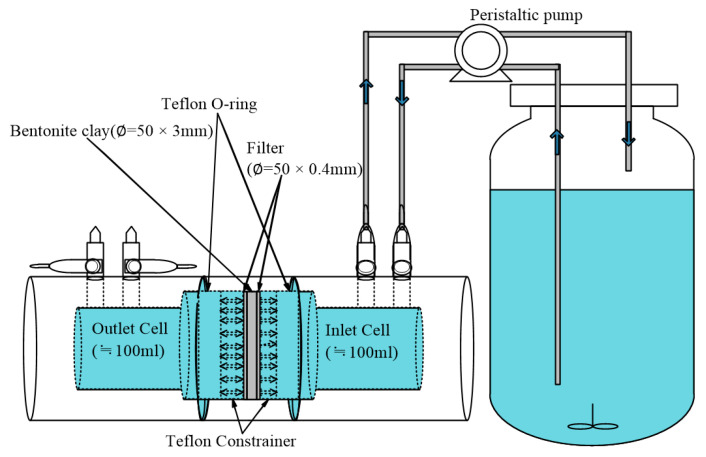
The TD apparatus with a sandwich-like column.

**Figure 2 materials-14-07056-f002:**
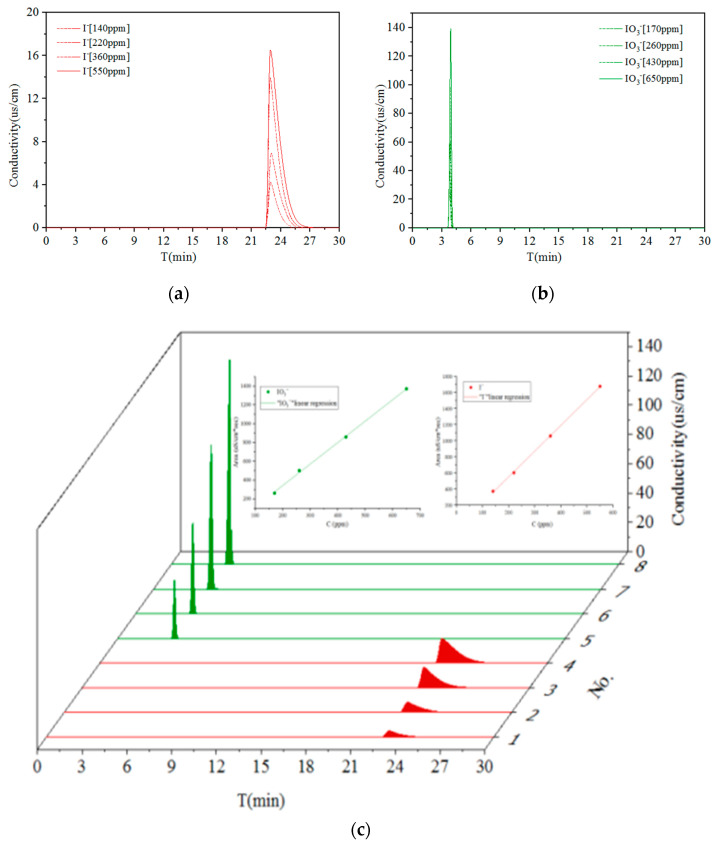
The chromatographic peaks for individual I^−^ and IO_3_^−^ at different concentrations. (**a**) I^−^; (**b**) IO_3_^−^; (**c**) the linear relationship between peak area and concentration of I^−^ and IO_3_^−^.

**Figure 3 materials-14-07056-f003:**
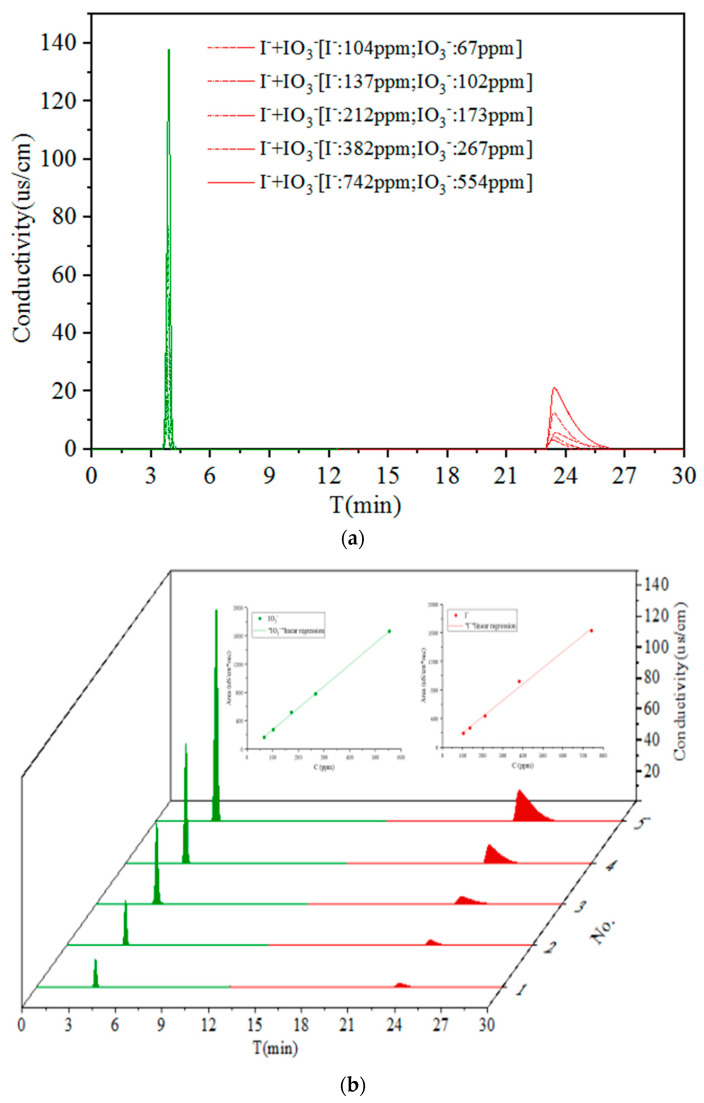
The chromatographical peak for co-existing I^−^ and IO_3_^−^ at different concentration. (**a**) I^−^ (green line) and IO_3_^−^ (red line) same in Figure 2; (**b**) the linear relationship between peak area and concentration of co-existing I^−^ and IO_3_^−^.

**Figure 4 materials-14-07056-f004:**
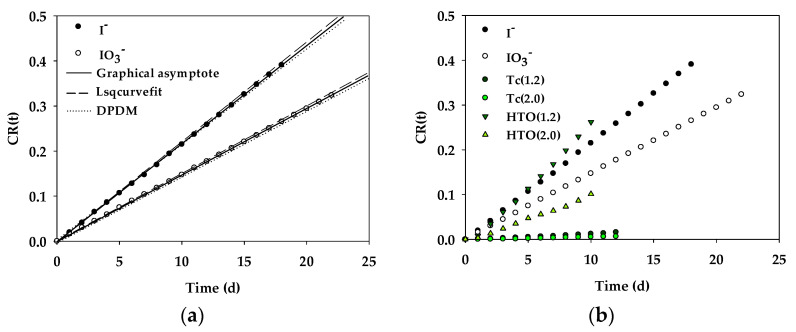
The diffusion curves of I^−^ and IO_3_^−^ in bentonite clay. (**a**) I^−^ and IO_3_^−^; (**b**) other radionuclides.

**Table 1 materials-14-07056-t001:** The chemical properties of commercial bentonite clay from three sources.

Element (%)	Wyoming	Taiwan	Inner Mongolia
SiO_2_	53.78	50.87	63.71
Al_2_O_3_	17.59	15.54	13.44
Fe_2_O_3_	3.24	5.93	2.01
CaO	1.29	2.83	0.90
Na_2_O	1.92	1.21	1.67
K_2_O	0.47	1.41	0.67
MnO	0.01	0.10	0.04
MgO	2.04	2.11	2.67
TiO_2_	0.14	0.38	0.12
P_2_O_5_	0.05	0.07	0.03
* LOI	16.77	12.72	12.91

* LOI: Loss On Ignition.

**Table 2 materials-14-07056-t002:** The characteristics of the individual I^−^ and IO_3_^−^ in different concentration.

Iodine Speciation	No.	C (ppm)	Area (uS/cm·s)
I^−^ (Retention time:~20 min)	1	140	372.2
	2	220	602.5
	3	360	1060.1
	4	550	1677.1
* R^2^		0.99	
IO_3_^−^ (Retention time:~4 min)	5	170	261.7
	6	260	503.0
	7	430	859.0
	8	650	1372.0
* R^2^		0.99	

* R^2^: R-squares.

**Table 3 materials-14-07056-t003:** The characteristics of the co-existing I^−^ and IO_3_^−^ at different concentration.

	Item	I^−^ (ppm)	Area (uS/cm·s)	IO_3_^−^ (ppm)	Area (uS/cm·s)	* R
No.						
1		104	192.60	67	166.14	79.36
2		137	266.94	102	273.76	68.67
3		212	436.76	173	518.48	52.30
4		382	921.08	267	781.06	71.17
5		742	1631.97	554	1667.85	69.19
** R ^2^		0.99		0.99		-

* R (resolution value); ** R ^2^ (linear regression coefficient).

**Table 4 materials-14-07056-t004:** The diffusion characteristics of different radionuclides in compacted Wyoming bentonite.

RN	*ρ_b_* (g/cm^3^)	*D_e_* (m^2^/s)	α_acc_	Reference
I^−^	2.0	3.28 × 10^−11^	0.20	In this work In this work
IO_3_^−^	2.0	2.43 × 10^−11^	0.25	
^99^TcO_4_^−^	2.0	4.38 × 10^−13^	0.06	Lee, C.P. et al. 2021 [25] Lee, C.P. et al. 2021 [25] Lee, C.P. et al. 2021 [25] Lee, C.P. et al. 2021 [25]
	1.2	8.89 × 10^−12^	0.39	
HTO	2.0	2.16 × 10^−11^	0.29	
	1.2	6.25 × 10^−11^	0.56	
SeO_3_^2−^ (IV)	1.2	1.50 × 10^−11^	3.52	Kong, J. et al. 2021 [20]

**Table 5 materials-14-07056-t005:** The pH, Eh, and distribution coefficients (*K_d_*) for I^−^ and IO_3_^−^ in three samples of bentonite clay.

RN	C_0_ (ppm)	Wyoming			Taiwan			Inner Mongolia		
		pH	Eh (mV)	*K_d_* (mL/g)	pH	Eh (mV)	*K_d_* (mL/g)	pH	Eh (mV)	*K_d_* (mL/g)
I^−^	5	9.46 ± 0.13	251 ± 6	0.01 ± 0.00	7.23 ± 0.08	170 ± 1	~0	9.22 ± 0.03	221 ± 3	0.01 ± 0.00
	10	9.55 ± 0.07	116 ± 5	~0	7.94 ± 0.45	282 ± 7	0.03 ± 0.00	9.05 ± 0.27	186 ± 15	~0
	20	9.57 ± 0.06	241 ± 2	~0	8.53 ± 0.04	306 ± 4	~0	9.11 ± 0.16	211 ± 8	~0
IO_3_^−^	5	9.53 ± 0.04	243 ± 9	~0	8.24 ± 0.01	321 ± 9	0.02 ± 0.00	9.13 ± 0.14	273 ± 8	0.01 ± 0.00
	10	9.26 ± 0.33	243 ± 8	0.01 ± 0.00	8.05 ± 0.02	247 ± 17	0.01 ± 0.00	9.36 ± 0.03	233 ± 12	0.01 ± 0.00
	20	9.38 ± 0.02	187 ± 9	~0	8.07 ± 0.11	225 ± 3	~0	9.28 ± 0.22	197 ± 4	~0

## Data Availability

The data presented in this study are available on request from the corresponding author.

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
