# Peer review of "An Improved Speciation Method Combining IC with ICPOES and Its Application to Iodide and Iodate Diffusion Behavior in Compacted Bentonite Clay"

_materials, 2021, doi:10.3390/ma14227056_

Round 1

Reviewer 1 Report

The contribution is importand due to the  presented possibility to qualitatively and quantitatively analyze co-existing iodide (I−) and iodate (IO3−) ions at various concentrations and use this possibility to distinguish the diffusion characteristics of these anions in compacted bentonite.

Some clarifications and modifications may be recommended for publication in the journal:

It is not necessary to mention anion exclusion effect in the title of the article.

Throughout the article, it is necessary to follow the usual convention that variables and constants are written in italics not only in the formulas, but also in the text.

Line 28 – alfa is not accessible porosity

The introduction to the issue of radioactive waste disposal is not directly related to the topic of the article, I recommend radically shortening (lines 34-95). Line 58 – the vitrification in not planned for spent nuclear fuel

Line 107 - I do not recommend referring to the work [24], because the wide range De is misleading without specifying the conditions of the experiments and the way of their evaluation.

Line 191 – c and C in eq. (1)?

Line 192 – introduce „capacity factor“ here

Line 204 – It does not seem useful to present both equations (3) and (4), check that the one in which "alpha" is present is presented.

Line 209 – The "time-lag" method was probably used in the evaluation, this could be mentioned. I recommend proving the extent to which the required ideal boundary conditions were met in the experiments.

I also recommend discussing, perhaps elsewhere, the influence on the evaluation of the presence of separating filters. In the opponent's opinion, the elimination of the diffusion resistance of the filters would highlight the difference between the diffusion coefficients belonging to the two I species studied.

Line 279 – tritiated water is obviously presented as HTO.

Line 281/Figure 4.  – The presented dependencies on the left graph are confusing. The best fit using relation (3) would suffice.

Table 5. – What is the dimension of C0?

Author Response

Dear Reviewer

We are very grateful to you for all your kind assistance. Several sentence and correction have been modified according to your suggestion.

As for filter effect in estimating Diffusion coefficients (De), the previous literature (Yamaguchi, T. et al. 2007) reported that concentration gradient between inlet and outlet diffusion cells with filters has few differences in estimating effective diffusion coefficients. In this work, we applied two glass microfiber filters with pore size of 0.7 um (GF/F, Associated Design & Manufacturing Co., Alexandria VA, U.S), Teflon O-ring and constrainers to reduce cell deformation, minimize diffusion resistance, and resist swelling pressure induced by the bentonite clay following saturation with water. According to Crank (1975) 12.4.2 Series-parallel formula, we also evaluate and calculate the D in our diffusion experiment, and it showed the D has only 5 to 10% variation for I and IO3.

  • Series-parallel formula, L1&L3: Length of filter, L2: bentonite slab

D1

L1

D2

L2

D3

L3

(L1/D1)+(L2/D2)+(L3/D3) = L/D

L2/D2= L/D- L1/D1-L3/D3

D2=L2/(L/D- L1/D1-L3/D3)

D2≒(L2/L)*D , L>>L1 , L2≒L , so D2≒D

Best Regards

N-C Tien

Reviewer 2 Report

This manuscript describes a study on the diffusion of I- and IO3- in compacted bentonite.

The focus of the study, however, is on the development of a method to analyse I- and IO3- by combining anion chromatography and ICP-OES.

The title is misleading. Perhaps better to use a title reflecting the focus of the work.

The introduction is a very long introduction explaining merely the concept of deep geological disposal. It is recommended to remove this long explanation an to focus on the topic.

What is the motivation of this work, what are the open questions and what is the approach to answer these questions.

When giving the properties of the bentonites used, it is better to give the mineralogy instead of the chemical composition. The mineral composition is of much more relevance to diffusion of anions as the chemical composition.

It is important to give the chemical composition of the solutions used in order to evaluate the diffusion parameters. In the manuscript, there is no information on the chemical composition of the used solutions.

The diffusion set-up should be better explained. There are also filters to confine the samples, and perforated plates. It is known that filters (and perforated plates) might have an effect on the diffusion. Therefore, the authors should evaluate whether or not filters and perforated plates have an effect or not. In the manuscript, this item is not mentioned at all.

The authors claim that sorption equilibrium was reached after 7 days. How can they state this when they did no time series ? In the experimental part it was stated that they waited 7 days.

Anion chromatography was already used earlier to look at the speciation of Iodine in diffusion studies. Look at the paper of:

Glaus M.A., Müller, W., Van Loon, L.R. Diffusion of iodide and iodate through Opalinus Clay: Monitoring of the redox state using an anion chromatographic technique. Applied Geochemistry 23 (2008) 3612–3619.

The novelty of the manuscript w.r.t. diffusion I not given. All observations made are not new. Everything can already be found in the literature. In that sense, publication of the manuscript is questionable.

Author Response

Dear Reviewer

We are very grateful to you for all your kind assistance. Several sentence and correction has been modified according to your suggestion.

  1. For I and IO3 diffusion result in Glaus M.A. et al. (2008), there is a major difference in Opalinus Clay(OPA, Mont Terri, Switzerland) and bentonite clay. In fact, the design of a deep geological repository for spent fuel or high-level radioactive waste (HLW) generally adopts a "multi-barrier system". Generally, the underground facilities, waste containers, and buffer/backfill materials (e.g. bentonite clay) are called engineering barriers, and the surrounding geological formations are called natural barriers. The OPA is natural barriers (surrounding geological formations) but MX-80 bentonite is a candidate materials for buffer/backfill materials. Namely, the buffer/backfill materials filled in around the radioactive wastes will play a very important role in retarding the migration of radionuclides when groundwater intrude the deep geological repository.
  2. In Glaus M.A. et al. (2008), radio-isotope (I-125) was applied to test in diffusion experiment with anion chromatography (IC). In fact, there are a lot of limitation and regulation for radio-isotope experiments in universities, and we applied stable isotope (I-127) for I and IO3 in this work to qualitative and quantitative analyses with ICP-OES combination. Moreover, it also showed a good agreement with radio isotope(I-125) results and would be a good method to combine with other instruments (ICPMS, AMS, Nano-Sims ) to study Iodine speciation in near future.
  3. As for filter effect in estimating diffusion coefficients, the previous literature (Yamaguchi, T. et al. 2007) reported that concentration gradient between inlet and outlet diffusion cells with filters has few differences in estimating effective diffusion coefficients. In this work, we applied two glass microfiber filters with pore size of 0.7 um (GF/F, Associated Design & Manufacturing Co., Alexandria VA, U.S), Teflon O-ring and constrainers to reduce cell deformation, minimize diffusion resistance, and resist swelling pressure induced by the bentonite clay following saturation with water. According to Crank (1975) 12.4.2 Series-parallel formula, we also evaluate and calculate the D in our diffusion experiment, and it showed the D has only 5 to 10% variation for I and IO3.
  • Series-parallel formula, L1&L3: Length of filter, L2: bentonite slab

D1

L1

D2

L2

D3

L3

(L1/D1)+(L2/D2)+(L3/D3) = L/D

L2/D2= L/D- L1/D1-L3/D3

D2=L2/(L/D- L1/D1-L3/D3)

D2≒(L2/L)*D , L>>L1 , L2≒L , so D2≒D

  1. In 2.1, XRD spectra for MX-80 bentonite has been published in previous works, it used to cite it in this paper. According to the database, montmorillonite is the major component and minor content with quartz and felspar. The DIW (de-ionized water) was applied in our diffusion experiment in order to qualitative and quantitative analyses with IC-ICP-OES combination.

Best Regards

N-C Tien

Reviewer 3 Report

This is an interesting work in which the authors implement a method to effectively analyse the presence of individual and coexisting I- and IO3-. The technique is based on a combination of ion chromatography (IC) and inductively coupled plasma optical emission spectrometry.  The authors find that the method is able to identify those ions. The work is relevant in the framework of the assessment of radioactive waste disposal facilities.

I think that the article can be accepted almost in its current form. However, before the final acceptance of the article, I would suggest the application of very few changes:

  • Very low resolution in Figure 1. Please change it since it is difficult to read the text.
  • Please, correct Table 3 (Item, No.) and introduce more significant decimals in the regression coefficient.
  • Please, introduce the lacking references in Table 4.

Author Response

Dear Reviewer

We are very grateful to you for all your kind assistance. Several sentence and correction in our manuscript have been modified according to your suggestion. Thanks again.

Best Regards

N-C Tien

Round 2

Reviewer 2 Report

Although the authors did some changes to the manuscript, this does not change the fact that the work is not novel and does not bring a better understanding of the diffusion of anions (or better I and IO3) in bentonite.

Further, there is still no information on the mineral composition of the bentonite and no information on the chemical composition of the pore water used. Both type information are important issues for understanding the diffusive behaviour of anions in bentonite.

I think that the authors intention is more on the technique they used to discriminate between I and IO3 and not on the diffusive behaviour. I recommend to change the title in this direction and to focus more on the analytical techniques than on the diffusive behaviour because there is really no added value w.r.t. the diffusive behaviour of anions in bentonite.

Author Response

Dear Reviewer

We are very grateful to you for all your kind assistance. The topic of our manuscript has been modified to “An improved speciation method combining by IC with ICPOES and its application to iodide and iodate diffusion behavior in compacted bentonite clay.”

The Wyoming MX-80 Na-bentonite clay was a commercial product and purchased from American Colloid Company. The basic characteristics have been reported for Its major minerals of approximately 85–90 wt% in previous works. Previous studies in our references [1,3,4,7,8, 23,24, 34] have investigated its mineralogical and chemical properties for porewater chemistry, and in this work diffusion of iodide and iodate in a saturation compacted clay (porous media) after water saturation (description in 2.4). During water saturation, the water in both cells has been moved out and refilled. Therefore, we hope the analytical method would be applied not only for Iodine but also in any oxi-reducing sensitive radionuclide(anion), such Se (SeO32- and SeO4 2-).    

Best Regards

N-C Tien
